# Monitoring the Spatiotemporal Difference in Glacier Elevation on Bogda Mountain from 2000 to 2017

**DOI:** 10.3390/ijerph18126374

**Published:** 2021-06-12

**Authors:** Weibing Du, Ningke Shi, Linjuan Xu, Shiqiong Zhang, Dandan Ma, Shuangting Wang

**Affiliations:** 1School of Surveying and Land Information Engineering, Henan Polytechnic University, Jiaozuo 454003, China; dwb@hpu.edu.cn (W.D.); zsqxcxy@163.com (S.Z.); my20001419@163.com (D.M.); wst@hpu.edu.cn (S.W.); 2State Key Laboratory of Desert and Oasis Ecology, Xinjiang Institute of Ecology and Geography, Chinese Academy of Sciences, Urumqi 830011, China; 3Yellow River Institute of Hydraulic Research, Yellow River Conservancy Commission of the Ministry of Water Resources, Zhengzhou 450003, China

**Keywords:** Sentinel-1A data, Bogda Mountain, glacier surface elevation, lake surface elevation

## Abstract

The difference in glacier surface elevation is a sensitive indicator of climate change and is also important for disaster warning and water supply. In this paper, 25 glaciers on Bogda Mountain, in the eastern Tianshan Mountains, are selected as the study object as they are typical of glaciers in arid or semi-arid areas with importance for water supply. The Repeat Orbit Interferometry (ROI) method is used to survey the surface elevation of these glaciers using Sentinel-1A Radar data from 2017. Using data from the Shuttle Radar Topography Mission (SRTM) and a Digital Elevation Model (DEM), the difference in the glacier surface elevation between 2000 and 2017 is obtained. A scheme to evaluate the accuracy of estimated variations in glacier surface elevation is proposed in this article. By considering the surfaces of lakes in the study region as ideal horizontal planes, the average standard deviation (SD) value of the lake elevation is taken as the error caused by the radar sensor and observing conditions. The SD of the lake elevation is used as an index to evaluate the error in the estimated variation of the glacier surface elevation, and the obtained SD values indicate that the result obtained using the ROI method is reliable. Additionally, the glacier surface elevation variation pattern and a Logarithmic Fitting Model (LFM) are used to reduce the error in high-altitude glacial accumulation areas to improve the estimation of the difference in the glacier surface elevation obtained using ROI. The average SD of the elevation of the 12 lakes is ±2.87 m, which shows that the obtained glacier surface elevations are reliable. This article concludes that, between 2000 and 2017, the surface elevation of glaciers on Bogda Mountain decreased by an average of 11.6 ± 1.3 m, corresponding to an average decrease rate of 0.68 m/a, and glaciers volume decreased by an average of 0.504 km^3^. Meanwhile, the surface elevations of the lakes increased by an average of 8.16 m. The decrease of glacier surface elevation leads to the expansion of glacial lakes. From the north slope clockwise to the south slope, the glacier elevation variation showed a decreasing trend, and the elevation variation gradually increased from the south slope to the north slope. With the increase of glacier altitude, the variation of glacier surface elevation gradually changed from negative to positive. The findings of this article suggest that the rate of glacier retreat on Bogda Mountain increased from 2000 to 2017.

## 1. Introduction

Glacier variation is a sensitive indicator of climate change [1] and is also important for disaster warning and water supply in arid and semi-arid areas [2]. Bogda glacier is a typical glacier with important water supply significance in arid and semi-arid areas [3]. The melting of glaciers in the Tianshan area has an important influence on the freshwater resources and economic development in Xinjiang and the whole of China. The glaciers in the east Tianshan mountains are very sensitive to climate change in Western China and their retreat and response to climate play an important indicating role. The monitoring of glaciers can identify their change characteristics, including their response to climate change. Moreover, the change of glacier melting and accumulation have an important impact on rivers and therefore also on ecosystems and economic production [4]. Thus, the advance and retreat of glaciers is of great significance to human activities and economic construction [5,6]. Additionally, the rapid melting of glaciers makes some lakes unstable, which leads to a series of ecological and environmental problems such as ice lake breakwaters, glacial floods, and changes in the coastal ecological environment, thereby affecting people’s health. Therefore, glacier monitoring is of great significance [7]. Glacier monitoring includes glacier area monitoring, height monitoring, volume monitoring, and mass balance monitoring. Among these, the monitoring of glacier elevation can reflect the volume change and elevation change of glaciers [8] and also provides support for the study of glacier mass balance (GMB).

The routine monitoring of changes in glacier elevation mainly involves field stereophotogrammetry, the Differential Global Positioning System technique, and ground-penetrating radar analysis [9,10]. However, although these methods have high precision, they can only be applied over a limited area [11]. Furthermore, the harsh environmental conditions in high-altitude areas can make the onsite measurement of glaciers dangerous, which represents another limitation of such measurements [12].

In recent years, with the rapid development of remote sensing technology, an increasing number of satellites that can be used for glacier elevation monitoring have become available and new elevation monitoring methods have been developed. Stereo mapping using visible-wavelength images, Interferometric Synthetic-aperture Radar (InSAR), satellite gravimetry, and laser altimetry are the main methods for glacier elevation monitoring. Moholdt et al. obtained recent and long-term elevation changes of Svalbard glacier based on laser altimetry measurements from ICESat [13,14]. Farintotti et al. used three independent methods based on satellite gravity measurements, laser altimetry, and glaciological modeling to estimate the change of the area of all glaciers in the Tianshan Mountains [15]. Furthermore, ICESat laser altimetry data and a Shuttle Radar Topography Mission (SRTM) Digital Elevation Model (DEM) were used to study the GMB in the Qilian Mountains, the Qinghai-Tibet Plateau, the Himalayans, and the Tianshan Mountains [16,17]. Some studies using the Repeat Orbit Interferometry (ROI) method estimated changes in GMB in the Tianshan Mountains using TanDEM-X SAR data to obtain a high-accuracy DEM [18]. These applications have shown that the ROI method using SAR data can provide elevation variation information with a high spatiotemporal resolution, thus providing comprehensive information about glacier morphological changes [19]. However, remote sensing methods have large uncertainties, and much work is needed to remove errors associated with remote sensing observations. In order to obtain reliable results for glacier elevation change, Farinotti [20] et al. used a simple dynamic model to obtain the spatially distributed thickness of individual glaciers by inverting their surface topography. The results were validated against a comprehensive set of thickness observations for 300 glaciers from most of the glaciated regions of the world. Huang et al. [21] proposed a new method based on the polynomial fitting of the GLAS footprint of the Sparse ICESat—GLAS to estimate the elevation change of the surface of two glaciers in Naimona’nyi glacier and Yanong glacier; the results showed that the polynomial fitting method gives a reliable estimation of elevation change. These methods offer new methods to monitor the elevation of glaciers on Bogda Mountain.

Researchers have invested large amounts of financial and human resources in monitoring the elevations of glaciers on Bogda Mountain [22]. For example, in the early 1980s, a joint Chinese–Japanese team estimated the accumulation, ablation characteristics, and material balance of the Sigonghe glacier No. 4. Additionally, researchers at the Tianshan Glacier Station conducted an extensive scientific investigation of a glacier on Bogda Mountain [23,24]. In this paper, Sentinel-1A radar data and Landsat images are used to obtain the surface elevations of glaciers on Bogda Mountain using the ROI method; the terrain law analysis of glacial central line was used to improve the accuracy of InSAR observations; the standard deviation (SD) of glacial lake elevation was used to evaluate the InSAR observational errors; the difference in the surface elevations of glaciers and glacial lakes was determined; and the response of glaciers to climate change was discussed.

## 2. Data and Methods 

### 2.1. Study Area 

Bogda Mountain, the highest peak in the eastern Tianshan Mountains, is located at 43.8° N, 88.3° E, around 50 km from Urumqi City, the capital of Xinjiang Province. The mountain has an average altitude of 5445 m. The location of the study area is shown in Figure 1. Prior to 1983, there were 113 glaciers on Bogda Mountain. However, this number has now decreased to 110 [25]. Many of these glaciers are of the valley type, such as the Heigou glacier No. 8 and the Sigonghe glacier No. 4, which are the sources of many rivers [26]. The Bogda Mountain area is cold and dry in winter and is influenced by westerly circulation in summer, which gives the area a continental climate typical of the mid-latitude westerlies. The annual and daily temperature varies greatly. In summer, precipitation is low, evaporation is high, and the variation of river runoff depends on glacial melting to a great extent [27]. 

As shown in Figure 1c, 25 typical glaciers on Bogda Mountain were investigated in this study. These include Heigou glacier No. 8 (which in this article is denoted Glacier No. 17); Sigonghe glacier No. 4, (here, Glacier No. 11); and fan-shaped distributary glaciers (here, glacier nos. 13 and 14). The elevations of these 25 glaciers were extracted using 25 glacial vector boundaries by using the vector file of CGI2 (the Second China Glacier Inventory) as a mask file. Additionally, as shown in Figure 1c, 12 lakes were also selected as study objects. Except for Lake 7, which is far from the glacier area, the rest of the lakes are within 100 m of a glacier and are closely related to the variation of glacier elevation. More information about the glaciers is shown in Table 1.

### 2.2. Data

The Sentinel-1A data were downloaded from the website of the European Space Agency (https://scihub.copernicus.eu/dhus, last access: 10 June 2021). To ensure the satisfactory coherence of the complex image pair, two complex SAR image pairs with intervals of 12 days were selected, as well as precise orbit data corresponding to SAR images [28]. Sentinel-1A is in a sun-synchronous orbit with an orbit height of 693 km and an inclination angle of 98.15°, and the introduction of precise orbit data can remove the systematic errors caused by orbit errors [29]. Table 2 shows the relevant information of the SAR image pairs that were used in this study to generate the DEM using the ROI method.

The elevation datum version 4 of the SRTM 3 DEM data is the geoid of EGM96 (Earth Gravitational Model 1996), and the plane datum is WGS84 (World Geodetic System 1984). The SRTM is a global DEM obtained using InSAR and has a vertical accuracy of ±16 m [30]. Additionally, the locations, names, sizes, and other general information of the glaciers on Bogda Mountain were obtained from CGI2. The CGI2 glacier information is based on remote sensing images and is compiled using a glacial delineation method and GIS technology [31]. A Landsat-8 image from 29 September 2017 and a Landsat-5 image from 2 September 2000 were selected to delineate lake boundaries and facilitate the analysis of lake elevation obtained from ROI.

This paper used temperature and precipitation data from 2000 to 2017 recorded at three meteorological stations near the study area, namely Dabancheng (Station No. 51477; 43.21° N, 88.19° E; altitude 1103.5 m), Kumish (Station No. 51526; 42.14° N, 88.13° E; altitude 922.4 m), and Turpan (Station No. 51573; 42.56° N, 89.12° E; altitude 34.5 m); these data were obtained from the China Meteorological Data Network (http://data.cma.cn/user/info.html, last access: 10 June 2021).

Synthetic-aperture radar (SAR) data acquired in summer in vertical–vertical polarization mode were selected. Summer images were chosen to avoid snow cover in non-glaciated areas, which would lead to serious decoherence in these areas. A SAR image pair for 2018 (see Table 2) was compared with a SAR image pair for 2017 (see Table 2). It was thus found that the accuracy of the 2017 pair was better than that of the 2018 pair; therefore, the SAR image pair for 2017 was chosen as a reference to study the difference in glacier elevation between 2000 and 2017. 

The central flow of data processing and results analysis is shown in Figure 2.

### 2.3. Methods

#### 2.3.1. Using the ROI Method to Generate a DEM

The ROI method was employed to compute the variation in glacier surface elevation using the SARscape software (Sarmap, Caslano, Switzerland). This method includes baseline estimation, interferogram generation, filtering, coherence calculation, phase unwrapping, track refinement, repeated removal of the phase of the flat earth, phase-to-elevation, and geocoding [32]. Following the above processes, using the DEM generated with the ROI method, the elevation of each pixel is calculated. Thus, the elevation of the glaciers in the study area is obtained. The glacier vector boundaries of CGI2 and glacier boundaries delineated from Landsat OLI images were used to clip the DEM image so that only glaciers and lakes were obtained [33]. Finally, the elevations of the 25 glaciers obtained from Sentinel-1A data were compared with the glacier elevations obtained from the SRTM DEM to obtain the glacier surface variation between 2000 and 2017. 

This study used an accuracy index based on the principle of interference theory. The ROI process is affected by many factors, such as snow scattering characteristics, decoherence effects, and radar imaging characteristics. It is difficult to quantify the influence of each factor separately. Therefore, in this study, a formula for the calculation of elevation accuracy [34] based on a large amount of practical modeling experience was used.
(1)$$pre=\sqrt{\frac{1-\gamma^{2}}{2\gamma^{2}}}\frac{\lambda R\sin\theta }{4\pi B}$$
where γ is the coherence coefficient, λ is the wavelength, *R* is the slant distance, *θ* is the local incidence angle, and *B* is the baseline length of the two radar images. However, further analysis is needed to assess the accuracy of the calculated elevation variation; this is because the backscatter of the radar signal of water is weak, which means that, in glacial and lake areas, the DEM accuracy is relatively low. In this study, in Section 2.3.2, a Logarithmic Fitting Model (LFM) was introduced to improve the accuracy of the DEM in glacial and lake areas.

To evaluate the calculated variation of glacier elevation, the SD of lake elevation was used as an index to measure the error of the elevation change of nearby glaciers. Each lake was extracted from Landsat 8 OLI images using the Normalized Difference Water Index (NDWI) [35]. Because the lake surface is a plane, the theoretical lake surface elevation is a unified value. However, due to the measurement error and the scattering characteristics of the lake itself, the lake surface elevation varies slightly. Assuming that the lake surface is an ideal horizontal plane [36], the SD value shown in Formula (2) was used as the error due to the radar sensor and observing environment.
(2)$$SD=\pm \sqrt{\frac{1}{N}\sum_{i=1}^{N}\left(x_{i}-u\right)}$$
where *N* is the number of pixels occupied by the lake, $x_{i}$ is the elevation value of pixel i, and u is the average of the elevation values of all the lake pixels. It was found that the average *SD* of the elevations of the 12 lakes was ±2.87 m, which is far smaller than the calculated variation of glacial elevation, suggesting that the obtained difference in glacier surface elevation is reliable. 

#### 2.3.2. Establishment of Logarithmic Fitting Model (LFM)

Here, the pixels whose glacier elevation variation values are within 1% of the actual variation are selected as the original sampling pixels. In Figure 3, Glacier No. 14 is selected as an example to illustrate the selection of the original sampling pixels. The selection rules were as follows: (1) avoid speckle areas (as can be seen in Figure 3); (2) select pixels located on a curve perpendicular to the isoline (as shown by the black sampling line in the enlarged view of Figure 3a); (3) select pixels where the elevation variation value decreases as the elevation increases. This was performed to make the original sampling pixels meet the prerequisite that the glacial accumulation area be located at a higher altitude and the glacial melting area be located at a lower altitude; and (4) select as many pixels as possible. The more pixels, the better the LFM analysis can reflect the real glacial elevation variation. The improved Glacier No. 14 elevation variation map after applying the LFM is shown in Figure 3b.

As shown in Figure 4, using the glacier elevation in 2000 as a reference point, the glacier elevation variation rule from 2000 to 2017 was determined by producing a scatter plot between the SRTM and the mean elevation variation, and subsequently, the LFM curve, which can reflect the characteristics of elevation variation, was fitted using these elevation variation points. Taking the value of the SRTM DEM as the independent variable and the elevation variation value of the corresponding pixel as the dependent variable, the LFM function reflects the change relationship between these two values. Finally, the SRTM DEM and the LFM function were used to calculate the glacier elevation change.

The glacier surface elevation variation [37] pattern is described as follows. For a glacier, normally, the GMB line can be defined as a contour line whose altitude is the average altitude of the whole glacier. The GMB line divides the glacier into two parts. Part 1 is the upper part, which represents the accumulation area with higher altitude; in this part, the glacier surface elevation increases because of the increased snowpack depth. If the glacier experiences accumulation, the maximum glacier surface elevation variation pixel is located in the accumulation part, and, as elevation decreases, the absolute elevation variation value reduces. Part 2 is the lower part, which represents the ablation area with lower altitude; in this part, the glacier surface elevation decreases because of the higher temperature. If the glacier retreats, the maximum glacier surface elevation variation pixel is located in the ablation part, and, as elevation increases, the absolute elevation variation value reduces. As shown in Figure 5, in the study area, between 2000 and 2017, the glacier variation trend was dominated by retreat. In the LFM, the dependent variable decreases as the independent variable increases, and the LFM was used to fit the elevation value and the elevation variation value. Here, the selected pixel takes the SRTM DEM value as the abscissa axis, and the elevation variation value of the corresponding position is the ordinate axis. Using the MATLAB software (MathWorks, Natick, MA, USA), the coordinates of the pixels were read and a scatter diagram was produced, as shown in Figure 4. From the scatter plot, it can be seen that the higher the elevation value, the higher the elevation variation value is and the larger the error trend. The LFM can reduce the elevation variation error of the glacier accumulation area, and the LFM result follows the glacier surface elevation variation rule. The MATLAB Curve Fitting Toolbox was used for fitting analysis, and the LFM was obtained using Formula (3).
(3)y=a*lnx+c
where *y* is the elevation variation value from 2000 to 2017, *x* is the elevation value for 2000 obtained from the SRTM DEM, and *a* and *c* are the fitting coefficients, with values of 117.3 and −982.5, respectively. Using the LFM, the coefficient of determination and the SD were found to be 0.907 and ±3.713 m, respectively. Because the coefficient of determination is close to 1 and the value of SD is not large, the LFM result is acceptable. The curve fitting is also shown in Figure 4. 

## 3. Results

### 3.1. Threshold Analysis of Glacier Elevation Variation

Due to the weak backscattering characteristics of water, the accuracy of DEMs produced using the ROI method is lower for glacial and lake areas. The error can reach hundreds of meters due to the complex radar scattering mechanism and environmental factors [38]. Therefore, in this study, a threshold of glacier elevation variation was determined and was taken as the maximum of the glacier elevation variation. This threshold was used as a reference to reduce the error. The results of previous studies of the elevation variation of glaciers on Bogda Mountain—which are summarized later in this paragraph—were used as a reference for estimating the maximum threshold for glacier variation. For example, in [39], researchers showed that the elevation of Heigou glacier No. 8 (Glacier No. 17 in Figure 6) decreased by 13 m between 1986 and 2009 and that the elevation of Sigonghe glacier No. 4 (Glacier No. 11 in Figure 6) decreased by 15 m between 1962 and 2009. The geographic locations of two fan-shaped distributary glaciers (glaciers No. 13 and No. 14 in Figure 6) and Sigonghe glacier No. 4 were similar, and their water and heat conditions were also similar; therefore, the elevations of the two glaciers decreased at similar rates from 1962 to 2006, reducing by nearly 15 m. 

Considering that the melting of glaciers may have accelerated in recent years, the threshold value of elevation variation was set to 1.5 times the experientially derived glacier elevation change introduced above, that is, the threshold is defined as ±25 m. Therefore, the elevation pixels whose absolute value of elevation variation is more than 25 m were removed, and the Kriging interpolation method was used to interpolate for the null-value pixels after this initial noise removal. The results of the estimated glacier surface elevation variation are shown in Figure 5. As shown in the figure, most of the glacial areas show elevation reduction in the ablation area with lower altitude under the GMB line and a slight elevation increase in the accumulation area with higher altitude above the GMB line. These observations are consistent with glacier dynamics theory [40]. However, the glacier surface elevation variation obtained in this way did not conform well to the variation obtained from CGI2. Therefore, the following improvements were made to improve the accuracy of the estimated glacier surface elevation variation.

### 3.2. LFM Results

Using the LFM and the elevation map for 2000 as the benchmark, the improved elevation variation results for the 25 glaciers and 12 lakes were obtained, as shown in Figure 6. From the above research, the following results can be obtained. Between 2000 and 2017, the glaciers on Bogda Mountain exhibited a retreating trend, and the glacier elevation decreased. During this period, the glacier volume decreased by an average of 0.504 km^3^, and the glacier elevation decreased by an average of 11.6 ± 1.3 m. In the following, the results obtained in this article are compared with previous results in the literature to prove the correctness of the proposed method.

(1) Glacier No. 17. The results of this study suggest that, from 2000 to 2017, the glacier’s volume decreased by 0.024 km^3^, and its average elevation decreased by 13 ± 0.9 m, representing an average decrease of 0.76 ± 0.05 m/a. This conclusion is consistent with the conclusions of a previous study [41], which found that, between 1986 and 2009, the volume of Glacier No. 17 decreased by 0.026 ± 0.0118 km^3^, the average thickness of the ice tongue decreased by 13 ± 6 m, and the thickness decreased by an average of 0.57 ± 0.26 m/a. Taking these results together with those obtained in the present study indicates that the thinning rate of Glacier No. 17 increased from 2000 to 2017, which indicates that the glacier retreat is becoming increasingly serious. (2) Glacier No. 11. The results of the present study suggest that, from 2000 to 2017, its volume decreased by 0.076 km^3^, and its average elevation decreased by 18 ± 1.9 m, representing an average decrease of 1.06 ± 0.11 m/a. This conclusion is consistent with the finding [42] that, between 1962 and 2009, the ice reserves of Glacier No. 11 decreased by 0.014 ± 0.0095 km^3^, the average thickness of the ice tongue decreased by 15 ± 10 m, and the thickness of the ice tongue decreased by an average of 0.32 ± 0.2 m/a. By comparing these results with those obtained in the present study, it can be seen that the thinning rate of Glacier No. 11 increased from 2000 to 2017 which indicates that the glacier is retreating faster. (3) Glacier No. 14. The results of this study indicate that, from 2000 to 2017, its volume decreased by 0.18 km^3^ and its average elevation decreased by 14 ± 1.1 m. Because the location of this glacier is similar to that of Glacier No. 11, and the water and heat conditions are similar, it has similar characteristics of elevation change. Therefore, from the above analysis, it can be seen that the estimated elevation variation of Glacier No. 14 is reliable. By comparing these results with the results of the present study, it can be found that the thinning rate of Glacier No. 14 increased from 2000 to 2017, which indicates that the glacier is retreating faster. The above comparison with previous literature results suggests that the results obtained in this paper are credible. The results of the elevation variations of other glaciers determined in the present study are shown in Figure 7. Figure 7a shows the maximum and minimum elevation variation of the 25 glaciers during the period between 2000 and 2017, and Figure 7b shows average elevation variation and SD.

### 3.3. Meteorological Data Processing Results

Using the annual precipitation anomaly and the annual average temperature change measured at the Dabancheng, Kumish, and Turpan meteorological stations in the area surrounding Bogda Mountain, a map showing the annual precipitation anomaly and annual average temperature change for 2000 to 2018 in the Bogda Mountain area was produced, as shown in Figure 8. From this figure, the following information can be obtained. From 2000 to 2018, the average temperature of the three meteorological stations was 11.07 °C and the average temperature increase rate was 0.42 °C·(10a)^−1^. The average precipitation was 50.6 mm and the decrease rate of precipitation was 8.75 mm·(10a)^−1^. Zhang Mingjun et al. found that for every 1 °C temperature rise, corresponding precipitation needs to increase by 47% or 56% to compensate for the melting of glaciers [43]. Therefore, the increase of precipitation cannot make up for the ablation caused by temperature rise. Using wavelet analysis, He et al. [44] showed that the Bogda Mountain area was in a period of high temperature and low precipitation from 1972 to 2013. Therefore, it can be inferred that temperature has a significant effect on glacier variation in the Bogda Mountain area.

## 4. Discussion

The variation of glacier surface elevation is related to many factors, such as glacier surface area, aspect, altitude, temperature, and precipitation. Therefore, based on the average elevation variation, this paper analyzes the relationship between the surface elevation change of glaciers in the Bogda Mountain area and the aforementioned five quantities.

### 4.1. Effects of Glacier Area on Glacier Elevation

By combining the average glacier elevation variation with the information of glacier area and altitude in CGI2, a relationship diagram was established between glacier area, glacier altitude, and average glacier elevation variation from 2000 to 2017, as shown in Figure 9. From the figure, it can be seen that most of the glaciers with large elevation variation are located at lower altitudes, while the glaciers with small elevation variation are located at higher altitudes. For glaciers of a similar size, glaciers with a lower altitude experienced more variation than glaciers with a higher altitude; that is, there is a relationship between glacier area variation and elevation. The results of this analysis suggest that, in general, the size of glaciers affects the overall elevation difference.

### 4.2. Effects of Glacier Aspect on Glacier Elevation

By combining the average glacier elevation variation with the information of glacier aspect in CGI2, the relationship between the average glacier elevation variation and aspect from 2000 to 2017 was established. The results are shown in Figure 10. The aspect is divided into eight areas, of which (0–22.5°, 337.5–360°) is north and (22.5–67.5°) is northeast. According to this classification rule, the clockwise direction is east, southeast, south, southwest, west, and northwest. According to the information shown in Figure 10, overall, the variation of the north slope aspect is relatively fast, and the variation of the south slope aspect is relatively slow. From the north slope clockwise to the south slope, the range of variation shows a decreasing trend and gradually expands from the south slope to the north slope. The rate of change is highest in the north direction and lowest in the south direction. This trend is consistent with the conclusions of Zhou Yuangang [45]. This phenomenon may be caused by the temperature difference caused by the influence of the aspect on the direction of water vapor movement and sunlight irradiance [46]. Additionally, because the studied glaciers are located at different altitudes, elevation is also an important factor causing the glacier surface elevation variation. In this paper, the relationship between glacier aspect and altitude from 2000 to 2017 was established, as shown in Figure 10 and Figure 11. It can be appreciated that the glaciers with fast ablation are mostly located at low altitudes, while glaciers with slow ablation or even accumulation are mostly in high-altitude areas. This proves that the influence of altitude on glacier elevation is greater than that of aspect on glacier variation.

### 4.3. Effects of Glacier Altitude on Glacier Elevation

By combining the average glacier elevation variation with the information of glacier altitude in CGI2, we established the relationship between the average elevation variation of glaciers and the altitude between 2000 and 2017, as shown in Figure 12. As shown in the figure, the elevation variations of the glaciers range from −25 to 20 m. Because the temperature rises faster in low-altitude areas than in high-altitude areas, glaciers at low altitudes experience more melting, so there is a decrease in the ablation area. With the increase of altitude, affected by snowfall, glaciers accumulate, thus the glacier elevation tends to increase in parts of the glaciers at higher altitude. With the increase of altitude, the glacier elevation variation gradually changes from negative to positive. The trend of glacier elevation difference is consistent with the glacier surface elevation variation pattern.

### 4.4. Effect of Lakes on Glacier Elevation

After a long period of melting, a lake will appear in the area before the glacier terminus. As shown in Figure 6, 11 lakes within 100 m of glaciers and one independent lake were selected. The basic information of these lakes is shown in Table 3. From Figure 6 and Table 3, it can be seen that Glacier No. 13 and Glacier No. 14 are closely related to Lake No. 1. Lake No. 7 is an independent natural lake, and the other lakes have a glacier meltwater supply. As shown in Table 3, the SD of the lake elevation variation was used as an index to evaluate the reliability of the glacier elevation variation. As shown in the table, the absolute SD values are all smaller than the lake elevation variation values and the glacier elevation variation values; this shows that the deformation information extracted using the ROI method is reliable. It was found that, for the lakes, there is a relationship between the lake area and the average glacier elevation variation. Between 2000 and 2017, the lake area expanded, the lake elevation increased, and correspondingly the glacier surface elevation decreased. This is due to the influence of glacial melting, which caused the glacial lake to expand and the lake’s average elevation to increase.

### 4.5. Effects of Meteorological Factors on Glacier Elevation

Glacier change is caused by climate change. Temperature and precipitation affect the development of glaciers and are the main climatic factors affecting glacier change; temperature determines the amount of melting and precipitation determines the amount of recharge [47,48,49]. Zhang Qibing et al. studied the relationship between glacier change and meteorological factors in the Qilian Mountains, an arid area in Western China. They found that glacier shrinkage is related to temperature rise [50]. In the present paper, through the processing of data from three meteorological stations around the Bogda Mountain area, we reached the conclusion that temperature has a significant effect on glacier variation in this area. The reliability of the results is indicated by the fact that Zhou et al. obtained the same results in their analysis of the influencing factors of glaciers in the Bogda peak area; that is, they found that the temperature has a significant effect on the change of the glacier area and snow cover area in this area [45]. Therefore, similar topography and environment may cause the same changes in glaciers.

## 5. Conclusions

In this paper, the surface elevation of Bogda Mountain in 2017 was obtained using orbit interferometry. Then, by comparison with the SRTM DEM of the mountain for 2000, the difference in the surface elevations of 25 glaciers on the mountain between 2000 and 2017 was obtained. It was found that the elevations of the glaciers decreased, and the glacier reserves decreased. From 2000 to 2017, the glacier volume decreased by an average of 0.504 km^3^, the elevation decreased by an average of 11.6 ± 1.3 m, and the average glacier elevation decrease rate was 0.68 m/a. As the elevations of the GMB lines of the glaciers rose, the glaciers tended to retreat faster. From 2000 to 2017, for the typical glaciers numbered Glacier No. 17, No. 11, and No. 14, the glacier volumes decreased by 0.024 km^3^, 0.076 m^3^, and 0.18 km^3^, the average glacier elevation decreased by 13 ± 0.9 m, 18 ± 1.9 m, and 14 ± 1.1 m, respectively, and the glaciers’ annual average elevation decreases were 0.76 ± 0.05 m/a, 1.06 ± 0.11 m/a, and 0.82 ± 0.064 m/a, respectively. These observations for these three glaciers are consistent with the conclusions of a previous study [41]. The comparison between the results presented in this study and previous results indicates that the glacial thinning rate increased from 2000 to 2017, which indicates that glacial retreat is becoming increasingly serious.

Finally, we analyzed the causes of the surface elevation variations of glaciers on Bogda Mountain from the aspects of glacier surface area, aspect, altitude, pre-glacial lake, temperature, and precipitation. The variations of glacier area, lake, and glacier elevation were small. From the north slope clockwise to the south slope, the glacier elevation variation showed a decreasing trend, and the elevation variation gradually increased from the south slope to the north slope. With the increase of glacier altitude, the variation of glacier surface elevation gradually changed from negative to positive. However, temperature has a significant effect on glaciers in the Bogda Mountain area.

The variation of glacier elevation is caused by many factors. Therefore, further studies are necessary to obtain more accurate estimates of glacier elevation changes.

## Figures and Tables

**Figure 1 ijerph-18-06374-f001:**
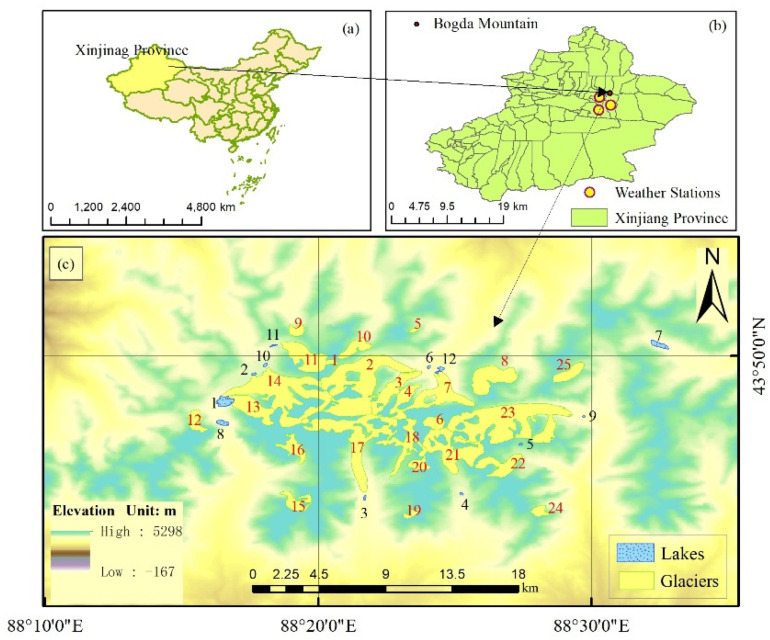
The research area of Bogda Mountain. (**a**) The administrative zoning map of China. (**b**) The administrative zoning map of Xinjiang Province. (**c**) A satellite image of Bogda Mountain in 2000. The red and black numbers show the reference numbers of the glaciers and lakes investigated in this study, respectively.

**Figure 2 ijerph-18-06374-f002:**
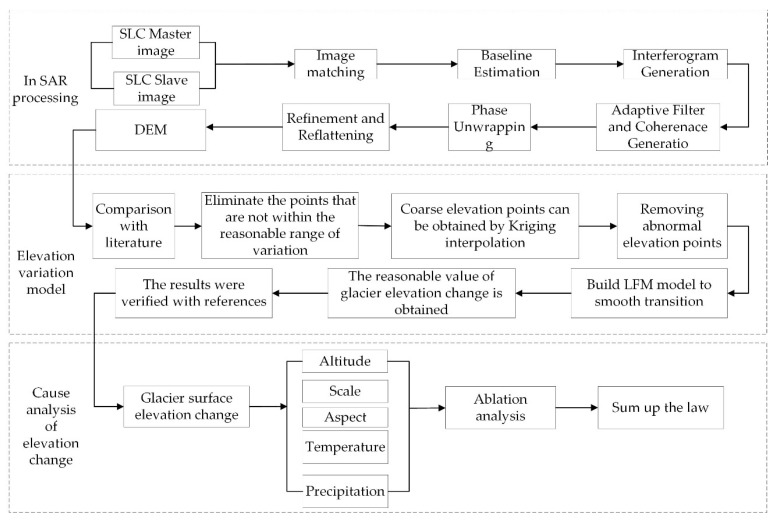
The central flow of data processing and results analysis.

**Figure 3 ijerph-18-06374-f003:**
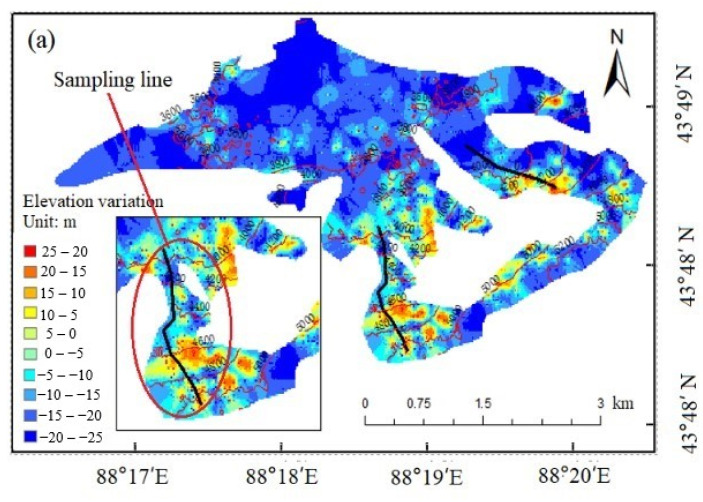
The elevation variation map of Glacier No. 14 before and after the application of the Logarithmic Fitting Model (LFM). (**a**) the elevation variation results obtained using the Repeat Orbit Interferometry (ROI) method. (**b**) the improved results by LFM.

**Figure 4 ijerph-18-06374-f004:**
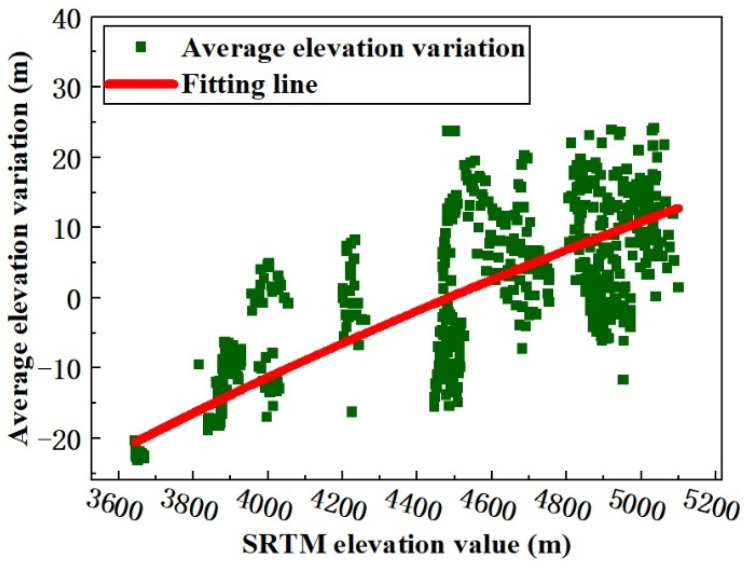
The LFM curve that was used to improve the ROI result. The green points are the sampling points and the red line is the fitting line obtained using the LFM.

**Figure 5 ijerph-18-06374-f005:**
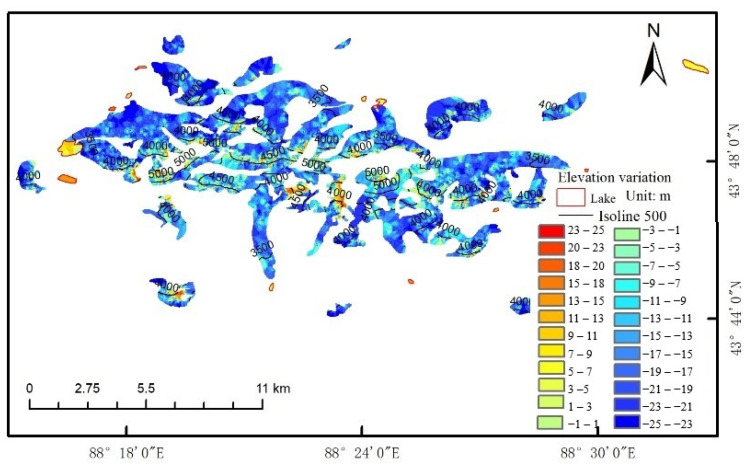
A map showing the change in elevation of glaciers on Bogda Mountain from 2000 to 2017. The red boundaries are the lake boundaries and the black lines are the isoline at an altitude of 500 m.

**Figure 6 ijerph-18-06374-f006:**
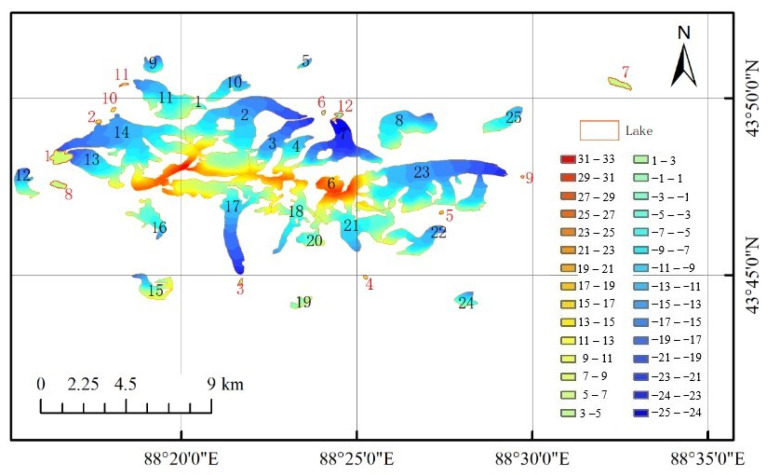
The elevation variation of the target glaciers from 2000 to 2017. The red boundaries are the lake boundaries.

**Figure 7 ijerph-18-06374-f007:**
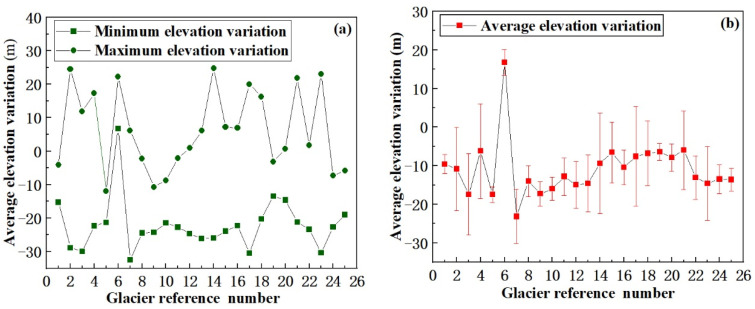
The elevation variation of the 25 studied glaciers in 2000–2017. (**a**) The maximum and minimum elevation variation. (**b**) The average elevation variation and SD.

**Figure 8 ijerph-18-06374-f008:**
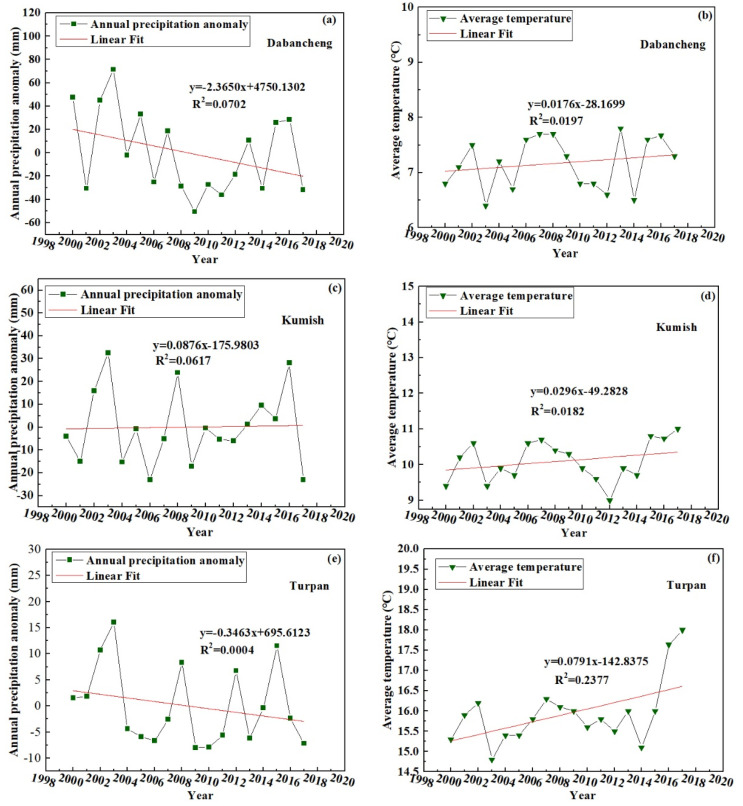
The annual average temperature and precipitation anomaly change in the Bogda Mountain area from 2000 to 2017. (**a**) The annual average precipitation anomaly change in the Dabancheng Station from 2000 to 2017. (**b**) The annual average temperature change in the Dabancheng Station from 2000 to 2017. (**c**) The annual average precipitation anomaly change in the Kumish Station from 2000 to 2017. (**d**) The annual average temperature change in the Kumish Station from 2000 to 2017. (**e**) The annual average precipitation anomaly change in the Turpan Station from 2000 to 2017. (**f**) The annual average temperature change in the Turpan Station from 2000 to 2017.

**Figure 9 ijerph-18-06374-f009:**
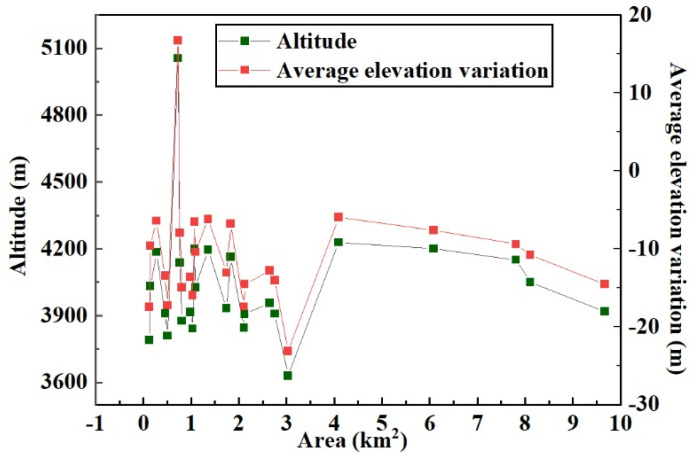
A graph showing the relationship between glacier area, glacier altitude, and average glacier elevation variation between 2000 and 2017.

**Figure 10 ijerph-18-06374-f010:**
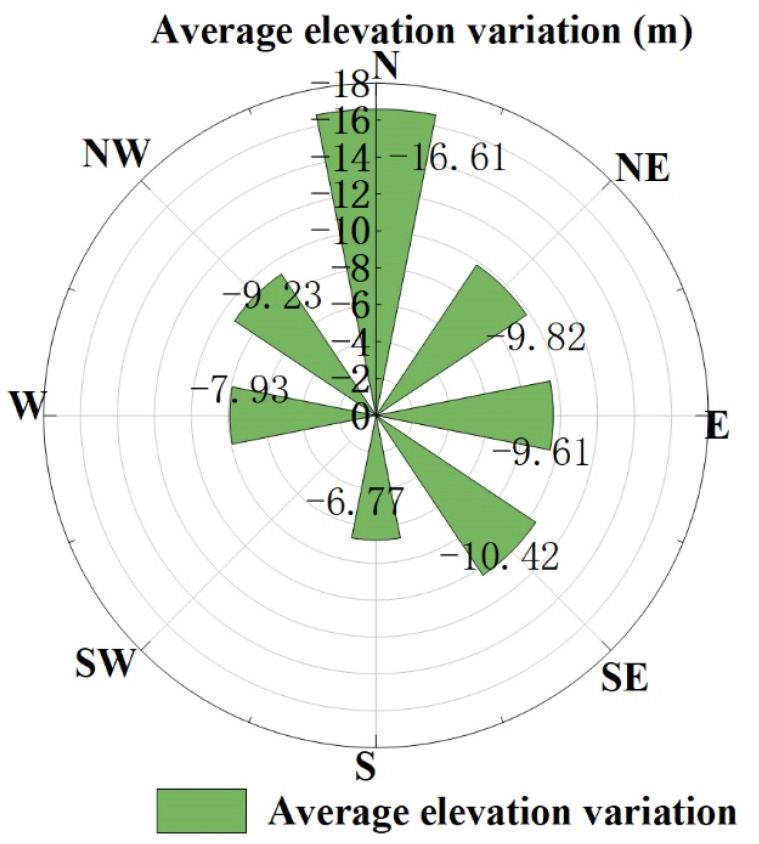
The relationship between glacier aspect and glacier altitude from 2000 to 2017.

**Figure 11 ijerph-18-06374-f011:**
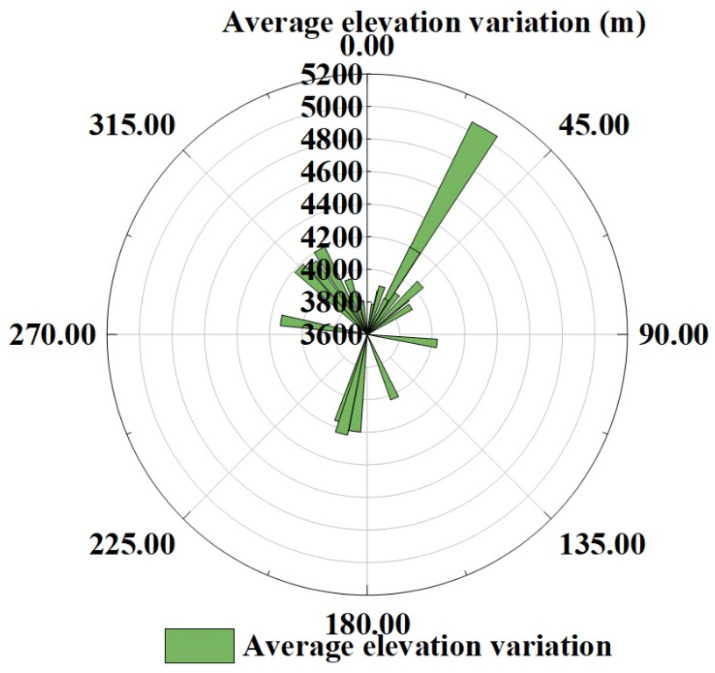
The relationship between glacier aspect and glacier elevation from 2000 to 2017.

**Figure 12 ijerph-18-06374-f012:**
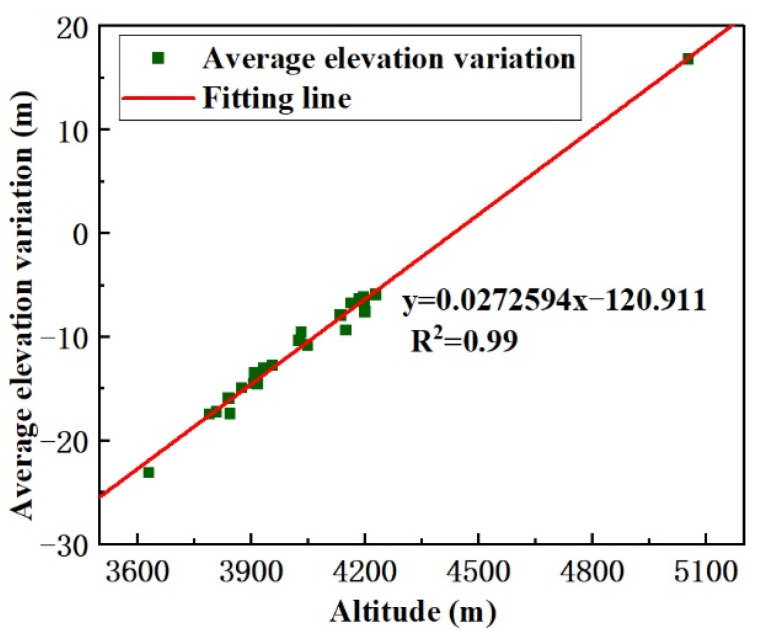
The relationship between the average elevation variation and altitude between 2000 and 2017. The green points are the average values for each glacier, and the red line is the fitting line that was used to study the relationships between the altitude and average elevation variation for each glacier.

**Table 1 ijerph-18-06374-t001:** Information about the glaciers used in the study.

Glacier No.	Glacier ID	Maximum Elevation (m)	Minimum Elevation (m)	Average Elevation (m)	Average Aspect (°)
1	5Y725B0010	4218.4	3846.9	4032.7	97.3
2	5Y725B0010	5393.7	3440.6	4049.7	46.7
3	5Y725B0010	5031.3	3405.2	3844.6	28.4
4	5Y725B0009	5096.1	3636.5	4196.7	29.5
5	5Y725B0011	3939.0	3670.6	3790.2	11.3
6	5Y725B0007	5304.5	4638.2	5055.2	29.8
7	5Y725B0008	4638.3	3344.7	3630.1	16.3
8	5Y725B0006	4319.3	3570.7	3908.5	17.2
9	5Y725C0007	4020.7	3592.9	3809.0	350.3
10	5Y725C0003	4067.2	3663.5	3840.9	29.2
11	5Y725D0004	4323.3	3657.4	3956.4	340.8
12	5Y812B0010	4381.9	3553.9	3875.8	16.2
12	5Y812B0011	4704.8	3548.1	3906.8	325.7
14	5Y725D0005	5428.6	3532.6	4150.7	322.0
15	5Y812B0022	4674.5	3604.1	4199.3	330.6
16	5Y812B0015	4634.1	3667.8	4027.3	156.7
17	5Y813B0008	5208.1	3401.4	4200.6	187.3
18	5Y813B0011	4994.0	3719.0	4164.0	197.2
19	5Y813B0015	4290.1	3955.1	4185.8	314.2
20	5Y813B0012	4421.7	3905.1	4136.6	279.3
21	5Y813B0017	5293.8	3684.4	4229.1	194.6
22	5Y813C0006	4469.0	3600.8	3933.5	47.6
23	5Y813C0010	5321.8	3400.4	3917.7	57.5
24	5Y813C0001	4140.4	3603.2	3909.4	36.5
25	5Y813C0012	4173.2	3730.9	3916.1	47.6

**Table 2 ijerph-18-06374-t002:** Information of the Sentinel-1A and Landsat satellite data that were used in this article.

SAR Date	Temporal Baseline	Spatial Baseline	Landsat Image Date	Sensor	Path/Row
29 August 2017	12 days	47.679 m	2 September 2000	TM	142/030
10 September 2017					
24 August 2018	12 days	100.872 m	29 September 2017	OLI	142/030
05 September 2018					

Note: SAR: Synthetic-aperture radar.

**Table 3 ijerph-18-06374-t003:** A comparison between the elevation variations of glaciers and lakes and the SD of the lake elevation variation.

Glacier No.	Lake No.	Lake Area (m^2^)	Lake Elevation Variation (m)	Glacier Elevation Variation (m)	SD of Lake Elevation (m)
1, 2, 3	6	41,600	7.09	−14.12	±0.5
7	12	136,000	4.15	−23.13	±0.08
11	11	42,000	3.26	−12.80	±0.004
12	8	248,800	15.71	−14.96	±0.9
17	3	50,400	7.77	−7.59	±0.04
21	4	36,400	16.32	−5.95	±0.00015
23	5	29,200	8.08	−14.56	±0.44
23	9	18,800	14.50	−14.56	±2.30
14	2	45,600	18	−9.40	±1.40
14	10	49,200	2.66	−9.40	±0.07
13	1	659,600	7.56	−10.81	±3.75
14	1	659,600	7.56	−17.43	±3.75
-	7	380,000	−6.52	-	±0.00042

## Data Availability

Not applicable.

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
