# Peer review of "Monitoring the Spatiotemporal Difference in Glacier Elevation on Bogda Mountain from 2000 to 2017"

_ijerph, 2021, doi:10.3390/ijerph18126374_

Round 1
Reviewer 1 Report
This manuscript presents a study to estimate surface elevation of glacier based on remote sensing data. This proposed work could benefit the monitoring of the change of glacier. Overall, I don’t find major defects, below are some minor issues: (1) The authors need to highlight the motivation of their study in introduction section, which is missing in current version. (2) The authors need to re-organize results and discussion sections. The authors shows many results in discussion section.Author Response
We would like to heartily thank you for the constructive comments. Your academic sense and scientific literacy definitely allowed us to improve the level of this manuscript. We highly appreciate your time and effort. Based on the constructive comments, careful modifications have been made to the manuscript. Please see the attachment for detailed responses.

Reviewer 2 Report
This paper is a study of how the elevation of glaciers has changed between 2000 and 2017. The study shows how the change in elevation over time is dependent on the location. It shows that at higher elevations the loss of ice is not as much as at the lower elevations. It is an important and interesting paper, given the interest in understanding effects of climate change and the need for monitoring changes and adapting to changes.
The authors provide a good survey of existing literature on this topic and give a reasonable motivation for the use of satellite remote sensing to accomplish the goal of monitoring glacier elevation change, given the hazardous environment to make in situ measurements, as well as the economics and scalability of such in-person measurements. Use of Landsat data for mapping lakes and the use of Sentnel-1 data for Repeat Orbit Interferometry is shown by the authors to be useful for monitoring elevation changes. Applying the techniques presented to Bogda mountains permits the authors to compare their results with previous results on a well-studied area.
The data and analysis technique employed are clearly explained. It is good that the authors provide information about the exact data that they used in their work (as in table 1). The results presented appear reasonable and physically meaningful.
Lines 149-151 – There is an incomplete sentence here that needs to be corrected.
Author Response
We would like to heartily thank you for the constructive comments. Your academic sense and scientific literacy definitely allowed us to improve the level of this manuscript. We highly appreciate your time and effort. Based on the constructive comments, careful modifications have been made to the manuscript. Please see the attachment for detailed responses.

Reviewer 3 Report
Review of the manuscript ID: ijerph-1213355 entitled: Monitoring the spatiotemporal variation of glacier elevation on Bogda Mountain from 2000 to 2017
This study analyses the surface elevation changes of 25 glaciers located on Mount Bogda, in the eastern Tianshan Mountains, joining the stream of studies on glacier volume changes, glacier mass balance, and mechanisms of these changes. The surface variability was investigated using repeat orbit interferometry (ROI) radar data (based on 2017 Sentinel-1A data) and SRTM elevation data, which were used to analyse the surface elevation changes of glacier DEMs from 2000 to 2017. The Authors used lake surfaces to determine the accuracy of the elevation change estimates, assuming that the lake sheets are ideal horizontal planes. The standard deviation of the lake surface elevation was used as an indicator to assess the error in the estimated glacier surface elevation variability. In addition, the Logarithmic Fit Model (LFM) was used to assess surface elevation changes. During the eight-year period analysed, the surfaces of the glaciers studied decreased by less than 12 m, corresponding to an average rate of decline of 0.68 m/a., and the volume of glaciers decreased by an average of 0.504 km3, while lake surface levels increased by about 8 m.
The proposed study is interesting, and its results also have a practical dimension, concerning the important issue of water resources and possibilities of their effective management. However, the manuscript has many shortcomings both in terms of concept, methodology and correctness of the English language.
Title: Both in the title and repeatedly in the text of the manuscript the Authors use the phrase "variation of glacier elevation". It is rather awkward. It would be better to use the phrase "difference of glacier surface elevation" referring to a technique for comparing surface changes developed for over 10 years called (DEM of Difference - DoD). Currently being developed as Geomorphic Change Detection Software (GCD). It would be appropriate to clean up the nomenclature and use 'elevation' as of a point above a relative ground point and 'altitude' as height above sea level (asl). In general, I would suggest to the Authors to use this very efficient tool (GCD) for elevation change assessment.
Abstract: Too much repetition. It should be shortened, but it should be supplemented with information on climatic conditions of described changes.
Introduction: Definitely too local character of references. Glacier surface change is a topic that is very widely referenced in the literature. In the introduction the references are mainly local. There is no information about the broad context of glacier research. The description of techniques is cursory. In lines 48-52, the authors refer to GPS techniques, however it is not only GPS, or rather now GNSS, also photogrammetric (SfM) and laser scanning (ALS, TLS) applications. This needs to be supplemented and give references to recent work e.g. from Svalbard, Iceland, Canadian Arctic, Alps, etc. In line 58 the Authors write 'many studies' and cite only 2 papers. Please be consistent.
Methodology: An important and rather poorly developed part of the paper. First, a flowchart with the stages of the study is missing. As a change comparison methodology I suggest using Geomorphic Change Detection Software (GCD), which allows simultaneous estimation of measurement uncertainty. The meteorological part is not clear to me. The description of the included range of meteorological measurements is cursory and it is not justified what it is supposed to be used for. The location of measurement stations relative to glaciers is not specified (add map). In general, this part raises my biggest doubts.
Results. The results lack the meteorological measurements that appear in the discussion as justification for glacier surface changes. This should be supplemented.
Discussion: The discussion is extensive but includes parts not justified in the description of the results (meteorological conditions). It is also local in nature, as is the introduction. There are only 3 references in the entire discussion (regional only). The Authors should relate their findings to changes described for other previously mentioned regions: Svalbard, Iceland, Canadian Arctic, Alps etc.
This study is useful, but needs refinement. Once the suggested changes are incorporated, it could be a valuable paper for readers of the International Journal of Environmental Research and Public Health.
Author Response

(The authors gave the same response as above.)

Reviewer 4 Report
See the attachment.

Author Response

(The authors gave the same response as above.)

Round 2
Reviewer 3 Report
Suggested changes were made by the Authors.
Author Response
Thanks for your suggestion. We greatly appreciate your time and effort. We would like to thank you for the valuable comments and constructive suggestions, which helped us significantly improve the value of the article.Please see the attachment for detailed responses.

Reviewer 4 Report
See the attachment

Author Response
We greatly appreciate your time and effort. We would like to thank the reviewer for the valuable comments and constructive suggestions, which helped us significantly improve the value of the article. The manuscript has been revised according to your comments.Please see the attachment for detailed responses.
